# Cooperative Graphical Models

**Josip Djolonga**
Dept. of Computer Science, ETH Zürich
josipd@inf.ethz.ch

**Stefanie Jegelka**
CSAIL, MIT
stefje@mit.edu

**Sebastian Tschiatschek**
Dept. of Computer Science, ETH Zürich
stschia@inf.ethz.ch

**Andreas Krause**
Dept. of Computer Science, ETH Zürich
krausea@inf.ethz.ch

## Abstract

We study a rich family of distributions that capture variable interactions significantly more expressive than those representable with low-treewidth or pairwise graphical models, or log-supermodular models. We call these *cooperative graphical models*. Yet, this family retains structure, which we carefully exploit for efficient inference techniques. Our algorithms combine the polyhedral structure of submodular functions in new ways with variational inference methods to obtain both lower and upper bounds on the partition function. While our *fully convex* upper bound is minimized as an SDP or via tree-reweighted belief propagation, our lower bound is tightened via belief propagation or mean-field algorithms. The resulting algorithms are easy to implement and, as our experiments show, effectively obtain good bounds and marginals for synthetic and real-world examples.

## 1 Introduction

Probabilistic inference in high-order discrete graphical models has been an ongoing computational challenge, and all existing methods rely on exploiting specific structure: either low-treewidth or pairwise graphical models, or functional properties of the distribution such as log-submodularity. Here, we aim to compute approximate marginal probabilities in complex models with long-range variable interactions that do not possess any of these properties. Instead, we exploit a combination of structural and functional properties in new ways.

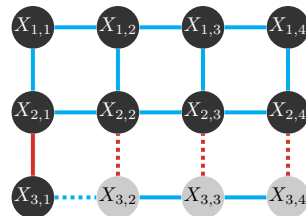

The classical example of image segmentation may serve to motivate our family of models: we would like to estimate a posterior marginal distribution over $k$ labels for each pixel in an image. A common approach uses Conditional Random Fields on a pixel neighborhood graph with *pairwise* potentials that encourage neighboring pixels to take on the same label. From the perspective of the graph, this model prefers configurations with few edges cut, where an edge is said to be cut if its endpoints have different labels. Such cut-based models,

Figure 1: Example cooperative model. Edge colors indicate the edge cluster. Dotted edges are cut under the current assignment.

however, short-cut elongated structures (e.g. tree branches), a problem known as *shrinking bias*. Jegelka and Bilmes [1] hence replace the bias towards short cuts (boundaries) by a bias towards configurations with certain higher-order structure: the cut edges occur at similar-looking pixel pairs. They group the graph *edges* into clusters (based on, say, color gradients across the endpoints), observing that the true object boundary is captured by few of these clusters. To encourage cutting edges from few clusters, the cost of cutting an edge *decreases* as more edges in its cluster are cut. In short, the edges "cooperate". In Figure 1, each pixel takes on one of two labels (colors), and cut

edges are indicated by dotted lines. The current configuration cuts three red edges and one blue edge, and has lower probability than the configuration that swaps $X_{3,1}$ to gray, cutting only red edges. Such a model can be implemented by an energy (cost) $h(\#\text{red edges cut}) + h(\#\text{blue edges cut})$, where e.g. $h(u) = \sqrt{u}$. Similar cooperative models can express a preference for shapes [2].

While being expressive, such models are computationally very challenging: the nonlinear function on *pairs* of variables (edges) is equivalent to a graphical model of extremely high order (up to the number of variables). Previous work hence addressed only MAP inference [3, 4]; the computation of marginals and partition functions was left as an open problem. In this paper, we close this gap, even for a larger family of models.

We address models, which we call *cooperative graphical models*, that are specified by an *undirected* graph $G = (V, E)$: each node $i \in V$ is associated with a random variable $X_i$ that takes values in $\mathcal{X} = \{1, 2, \ldots, k\}$. To each vertex $i \in V$ and edge $\{i, j\}$, we attach a potential function $\theta_i \colon \mathcal{X} \to \mathbb{R}$ and $\theta_{i,j} \colon \mathcal{X}^2 \to \mathbb{R}$, respectively. Our distribution is then

$$P(\mathbf{x}) = \frac{1}{\mathcal{Z}} \exp\left(-\left(\sum_{i \in V} \theta_i(x_i) + \sum_{\{i,j\} \in E} \theta_{i,j}(x_i, x_j) + f(\mathbf{y}(\mathbf{x}))\right)\right) \nu(\mathbf{x}), \qquad (1)$$

where we call $\mathbf{y} \colon \mathcal{X}^n \to \{0, 1\}^E$ the *disagreement variable*[1], defined as $y_{i,j} = [\![ x_i \neq x_j ]\!]$. The term $\nu \colon \mathcal{X}^n \to \mathbb{R}_{\geq 0}$ is the *base-measure* and allows to encode constraints, e.g., conditioning on some variables. With $f \equiv 0$ we obtain a Markov random field.

Probabilistic inference in our model class (1) is very challenging, since we make *no factorization assumption* about $f$. One solution would be to encode $P(\mathbf{x})$ as a log-linear model via a new variable $\mathbf{z} \in \{0, 1\}^E$ and constraints $\nu(\mathbf{x}, \mathbf{z}) = [\![ \mathbf{y}(\mathbf{x}) = \mathbf{z} ]\!]$, but this in general requires computing exponential-sized sufficient statistics from $\mathbf{z}$. In contrast, we make one additional key assumption that will enable the development of efficiently computable variational lower and upper bounds: we henceforth assume that $f \colon \{0, 1\}^E \to \mathbb{R}$ is *submodular*, i.e., it satisfies

$$f(\min(\mathbf{y}, \mathbf{y}')) + f(\max(\mathbf{y}, \mathbf{y}')) \leq f(\mathbf{y}) + f(\mathbf{y}') \quad \text{for all } \mathbf{y}, \mathbf{y}' \in \{0, 1\}^E,$$

where the $\min$ and $\max$ operations are taken element-wise. For example, the pairwise potentials $\theta_{i,j}$ are submodular if $\theta_{i,j}(0, 0) + \theta_{i,j}(1, 1) \leq \theta_{i,j}(0, 1) + \theta_{i,j}(1, 0)$. In our introductory example, $f$ is submodular if $h$ is concave. As opposed to [3], we do not assume that $f$ is monotone increasing. Importantly, even if $f$ is submodular, $P(\mathbf{x})$ neither has low treewdith, nor is its logarithm sub- or supermodular in $\mathbf{x}$, properties that have commonly been exploited for inference.

**Contributions.** We make the following contributions: (1) We introduce a new family of probabilistic models that can capture rich non-submodular interactions, while still admitting efficient inference. This family includes pairwise and certain higher-order graphical models, cooperative cuts [1], and other, new models. We develop new inference methods for these models; in particular, (2) upper bounds that are amenable to *convex* optimization, and (3) lower bounds that we optimize with traditional variational methods. Finally, we demonstrate the efficacy of our methods empirically.

## 1.1 Related work

**Maximum-a-posteriori (MAP).** Computing the mode of (1) for binary models is also known as the *cooperative cut* problem, and has been analyzed for the case when both the pairwise interactions $\theta_{i,j}$ are submodular and $f$ is monotone [1]. While the general problem is NP-hard, it can be solved if $f$ is defined by a piecewise linear concave function [4].

**Variational inference.** Since computing marginal probabilities for (1) is #P-hard even for pairwise models (when $f \equiv 0$) [5, 6], we revert to approximate inference. Variational inference methods for discrete pairwise models have been studied extensively; a comprehensive overview may be found in [7]. We will build on a selection of techniques that we discuss in the next section. Most existing methods focus on pairwise models ($f \equiv 0$), and many scale exponentially with the size of the largest factor, which is infeasible for our cooperative models. Some specialized tractable inference methods exist for higher-order models [8, 9], but they do not apply to our family of models (1).

**Log-supermodular models.** A related class of relatively tractable models are distributions $P(\mathbf{x}) = \frac{1}{Z}\exp(-g(\mathbf{x}))$ for some submodular function $g$; Djolonga and Krause [10] showed variational inference methods for those models. However, our models are not log-supermodular. While [10] also obtain upper and lower bounds, we need different optimization techniques, and also different polytopes. In fact, submodular and multi-class submodular [11] settings are a strict subset of ours: the function $g(x)$ can be expressed via an auxiliary variable $z \in \{0, 1\}$ that is fixed to zero using $\nu(\mathbf{x}, z) = [\![z = 0]\!]$. We then set $f(\mathbf{y}(\mathbf{x}, z)) = g(x_1 \neq z, x_2 \neq z, \dots, x_n \neq z)$.

## 2 Notation and Background

Throughout this paper, we have $n$ variables in a graph of $m$ edges, and the potentials $\theta_i$ and $\theta_{i,j}$ are stored in a vector $\boldsymbol{\theta}$. The characteristic vector (or indicator vector) $\mathbf{1}_A$ of a set $A$ is the binary vector which contains 1 in the positions corresponding to elements in $A$, and zeros elsewhere. Moreover, the vector of all ones is $\mathbf{1}$, and the neighbours of $i \in V$ are denoted by $\delta(i) \subseteq V$.

**Submodularity.** We assume that $f$ in Eqn. (1) is submodular. Occasionally (in Sec. 4 and 5, where stated), we assume that $f$ is *monotone*: for any $\mathbf{y}$ and $\mathbf{y}'$ in $\{0, 1\}^E$ such that $\mathbf{y} \leq \mathbf{y}'$ coordinate-wise, it holds that $f(\mathbf{y}) \leq f(\mathbf{y}')$. When defining the inference schemes, we make use of two polytopes associated with $f$. First, the *base polytope* of a submodular function $f$ is

$$B(f) = \{\mathbf{g} \in \mathbb{R}^m \mid \forall \mathbf{y} \in \{0, 1\}^E : \mathbf{g}^T\mathbf{y} \leq f(\mathbf{y})\} \cap \{\mathbf{g} \in \mathbb{R}^m \mid \mathbf{g}^T\mathbf{1} = f(\mathbf{1})\}.$$

Although $B(f)$ is defined by exponentially many inequalities, an influential result [12] states that it is tractable: we can optimize linear functions over $B(f)$ in time $O(m \log m + mF)$, where $F$ is the time complexity of evaluating $f$. This algorithm is part of our scheme in Figure 2. Moreover, as a result of this (linear) tractability, it is possible to compute orthogonal projections onto $B(f)$. Projection is equivalent to the *minimum norm point problem* [13]. While the general projection problem has a high degree polynomial time complexity, there are many very commonly used models that admit practically fast projections [14, 15, 16].

The second polytope is the *upper submodular polyhedron of $f$* [17], defined as

$$\mathcal{U}(f) = \{(\mathbf{g}, c) \in \mathbb{R}^{m+1} \mid \forall \mathbf{y} \in \{0, 1\}^E : \mathbf{g}^T\mathbf{y} + c \geq f(\mathbf{y})\}.$$

Unfortunately, $\mathcal{U}(f)$ is not as tractable as $B(f)$: even checking membership in $\mathcal{U}(f)$ is hard [17]. However, we can still succinctly describe specific elements of $\mathcal{U}(f)$. In §4, we show how to efficiently optimize over those elements.

**Variational inference.** We briefly summarize key results for variational inference for pairwise models, following Wainwright and Jordan [7]. We write pairwise models as[2]

$$P(\mathbf{x}) = \exp\left(-\left(\sum_{i \in V} \theta_i(x_i) + \sum_{\{i,j\} \in E} (g_{i,j}[\![x_i \neq x_j]\!] + \theta_{i,j}(x_i, x_j))\right) - A(\mathbf{g})\right)\nu(\mathbf{x}),$$

where $\mathbf{g} \in \mathbb{R}^E$ is an arbitrary vector and $A(\mathbf{g})$ is the log-partition function. For any choice of parameters $(\boldsymbol{\theta}, \mathbf{g})$, there is a resulting vector of marginals $\boldsymbol{\mu} \in [0, 1]^{k|V|+k^2|E|}$. Specifically, for every $i \in V$, $\boldsymbol{\mu}$ has $k$ elements $\mu_{i,x_i} = P(X_i = x_i)$, one for each $x_i \in \mathcal{X}$. Similarly, for each $\{i, j\} \in E$, there are $k^2$ elements $\mu_{ij,x_ix_j}$ so that $\mu_{ij,x_ix_j} = P(X_i = x_i, X_j = x_j)$. The marginal polytope $\mathbb{M}$ is now the set of all such vectors $\boldsymbol{\mu}$ that are realizable under *some* distribution $P(\mathbf{x})$, and the partition function can equally be expressed in terms of the marginals [7]:

$$A(\mathbf{g}) = \sup_{\boldsymbol{\mu} \in \mathbb{M}} \underbrace{\left(-\sum_{i \in V, x_i} \mu_{i,x_i}\theta_i(x_i) - \sum_{\{i,j\} \in E}\sum_{x_i,x_j} \mu_{ij,x_ix_j}\theta_{i,j}(x_i, x_j) - \boldsymbol{\Delta}(\boldsymbol{\mu})^T\mathbf{g}\right)}_{\langle \mathtt{stack}(\boldsymbol{\theta}, \mathbf{g}), \boldsymbol{\mu}\rangle} + H(\boldsymbol{\mu}), \quad (2)$$

where $H(\boldsymbol{\mu})$ is the entropy of the distribution, $\boldsymbol{\Delta}(\boldsymbol{\mu})$ is the vector of disagreement probabilities with entries $\Delta(\boldsymbol{\mu})_{i,j} = \sum_{x_i \neq x_j} \boldsymbol{\mu}_{ij,x_ix_j}$, and $\mathtt{stack}(\boldsymbol{\theta}, \mathbf{g})$ adds the elements of $\boldsymbol{\theta}$ and $\mathbf{g}$ into a single

vector so that the sum can be written as an inner product. Alas, neither $\mathbb{M}$ nor $H(\boldsymbol{\mu})$ have succinct descriptions and we will have to approximate them. Because the vectors in the approximation of $\mathbb{M}$ are in general not correct marginals, they are called *pseudo-marginals* and will be denoted by $\boldsymbol{\tau}$ instead of $\boldsymbol{\mu}$. Different approximations of $\mathbb{M}$ and $H$ yield various methods, e.g. mean-field [7], the semidefinite programming (SDP) relaxation of Wainwright and Jordan [18], tree-reweighted belief propagation (TRWBP) [19], or the family of weighted entropies [20, 21]. Due to the space constraints, we only discuss the latter. They approximate $\mathbb{M}$ with the *local polytope*

$$\mathbb{L} = \{\boldsymbol{\tau} \geq \mathbf{0} \mid (\forall i \in V) \sum_{x_i} \tau_{i,x_i} = 1 \text{ and } (\forall j \in \delta(i)) \, \tau_{i,x_i} = \sum_{x_j} \tau_{ij,x_i x_j}\}.$$

The approximations $\overline{H}$ to the entropy $H$ are parametrized by one weight $\rho_{i,j}$ per edge and one $\rho_i$ per vertex $i$, all collected in a vector $\boldsymbol{\rho} \in \mathbb{R}^{|V|+|E|}$. Then, they take the following form

$$\overline{H}(\boldsymbol{\tau}, \boldsymbol{\rho}) = \sum_{i \in V} \rho_i H_i(\boldsymbol{\tau}_i) + \sum_{\{i,j\} \in E} \rho_{i,j} H_{i,j}(\boldsymbol{\tau}_{i,j}), \text{where} \quad \begin{array}{ll} H_i(\boldsymbol{\tau}_i) & = -\sum_{x_i} \tau_{i,x_i} \log \tau_{i,x_i}, \text{ and} \\ H_{i,j}(\boldsymbol{\tau}_{i,j}) & = -\sum_{x_i,x_j} \tau_{ij,x_i j x_j} \log \tau_{ij,x_i x_j}. \end{array}$$

The most prominent example is traditional belief propagation, i.e., using the Bethe entropy, which sets $\rho_e = 1$ for all $e \in E$, and assigns to each vertex $i \in V$ a weight of $\rho_i = 1 - |\delta(i)|$.

## 3 Convex upper bounds

The above variational methods do not directly generalize to our cooperative models: the vectors of marginals could be exponentially large. Hence, we derive a different approach that relies on the submodularity of $f$. Our first step is to approximate $f(\mathbf{y}(\mathbf{x}))$ by a linear lower bound, $f(\mathbf{y}(\mathbf{x})) \approx \mathbf{g}^T \mathbf{y}(\mathbf{x})$, so that the resulting (pairwise) linearized model will have a partition function *upper bounding* that of the original model. Ensuring that $g$ indeed remains a lower bound means to satisfy an exponential number of constraints $f(\mathbf{y}(\mathbf{x})) \geq \mathbf{g}^T \mathbf{y}(\mathbf{x})$, one for each $\mathbf{x} \in \{0, 1\}^n$. While this is hard in general, the submodularity of $f$ implies that these constraints are easily satisfied if $\mathbf{g} \in B(f)$, a very tractable constraint. For $\mathbf{g} \in B(f)$, we have

$$\log \mathcal{Z} = \log \sum_{\mathbf{x} \in \{0,1\}^V} \exp \big( -(\sum_{i \in V} \sum_{x_i} \theta_i(x_i) + \sum_{\{i,j\} \in E} \theta_{i,j}(x_i, x_j) + f(\mathbf{y}(\mathbf{x}))) \big)$$

$$\leq \log \sum_{\mathbf{x} \in \{0,1\}^V} \exp \big( -(\sum_{i \in V} \sum_{x_i} \theta_i(x_i) + \sum_{\{i,j\} \in E} (\theta_{i,j}(x_i, x_j) + g_{i,j}[\![x_i \neq x_j]\!])) \big) \equiv A(\mathbf{g}).$$

Unfortunately, $A(\mathbf{g})$ is still very hard to compute and we need to approximate it. If we use an approximation $\overline{A}(\mathbf{g})$ that *upper bounds* $A(\mathbf{g})$, then the above inequality will still hold when we replace $A$ by $\overline{A}$. Such approximations can be obtained by relaxing the marginal polytope $\mathbb{M}$ to an *outer* bound $\overline{\mathbb{M}} \supseteq \mathbb{M}$, *and* using a concave entropy surrogate $\overline{H}$ that upper bounds the true entropy $H$. TRWBP [19] or the SDP formulation [18] implement this approach. Our central optimization problem is now to find the tightest upper bound, an optimization problem[3] in $\mathbf{g}$:

$$\underset{\mathbf{g} \in B(f)}{\text{minimize}} \sup_{\boldsymbol{\tau} \in \overline{\mathbb{M}}} \langle \mathtt{stack}(\boldsymbol{\theta}, \mathbf{g}), \boldsymbol{\tau} \rangle + \overline{H}(\boldsymbol{\tau}). \tag{3}$$

Because the inner problem is *linear* in $\mathbf{g}$, this is a *convex* optimization problem over the base polytope. To obtain the gradient with respect to $\mathbf{g}$ (equal to the negative disagreement probabilities $-\boldsymbol{\Delta}(\boldsymbol{\tau})$), we have to solve the inner problem. This subproblem corresponds to performing variational inference in a *pairwise* model, e.g. via TRWBP or an SDP. The optimization properties of the problem (3) depend on its *Lipschitz continuity* of the gradients (smoothness). Informally, the inferred pseudomarginals should not drastically change if we perturb the linearization $\mathbf{g}$. The formal condition is that there exists some $\sigma > 0$ so that $\|\boldsymbol{\Delta}(\boldsymbol{\tau}) - \boldsymbol{\Delta}(\boldsymbol{\tau}')\| \leq \sigma \|\boldsymbol{\tau} - \boldsymbol{\tau}'\|$ for all $\boldsymbol{\tau}, \boldsymbol{\tau}' \in \overline{\mathbb{M}}$. We discuss below when this condition holds. Before that, we discuss two different algorithms for solving problem (3), and how their convergence depends on $\sigma$.

**Frank-Wolfe.** Given that we can efficiently solve linear programs over $B(f)$, the Frank-Wolfe [23] algorithm is a natural candidate for solving the problem. We present it in Figure 2. It iteratively moves towards the minimizer of a linearization of the objective around the current iterate. The method has a convergence rate of $O(\sigma/t)$ [24], where $\sigma$ is the assumed smoothness parameter. One can either use a fixed step size $\gamma = 2/(t+2)$, or determine it using line search. In each iteration, the algorithm calls the procedure LINEAR-ORACLE, which finds the vector $\mathbf{s} \in B(f)$ that minimizes the linearization of the objective function in (3) over the base polytope $B(f)$. The linearization is given by the (approximate) gradient $\mathbf{\Delta}(\boldsymbol{\tau})$, determined by the computed approximate marginals $\boldsymbol{\tau}$.

When taking a step towards $\mathbf{s}$, the weight of edge $e_i$ is changed by $s_{e_i} = f(\{e_1, e_2, \ldots, e_i\}) - f(\{e_1, e_2, \ldots, e_{i-1}\})$. Due to the submodularity[4] of $f$, an edge will obtain a higher weight if it appears earlier in the order determined by the disagreement probabilities $\mathbf{\Delta}(\boldsymbol{\tau})$. Hence, in every iteration, the algorithm will re-adjusts the pairwise potentials, by encouraging the variables to agree more as a function of their (approximate) disagreement probability.

1: **procedure** FW-INFERENCE($f, \boldsymbol{\theta}$)
2:    $\mathbf{g} \leftarrow$ LINEAR-ORACLE($f, \mathbf{0}$)
3:    **for** $t = 0, 1, \ldots, \mathtt{max\_steps}$ **do**
4:       $\boldsymbol{\tau} \leftarrow$ VAR-INFERENCE($\boldsymbol{\theta}, \mathbf{g}$)
5:       $\mathbf{s} \leftarrow$ LINEAR-ORACLE($f, \boldsymbol{\tau}$)
6:       $\gamma \leftarrow$ COMPUTE-STEP-SIZE($\mathbf{g}, \mathbf{s}$)
7:       $\mathbf{g} \leftarrow (1-\gamma)\mathbf{g} + \gamma\mathbf{s}$
8:    **return** $\boldsymbol{\tau}, \hat{A}$

1: **procedure** LINEAR-ORACLE($f, \boldsymbol{\tau}$)
2:    Let $e_1, e_2, \ldots, e_{|E|}$ be the edges $E$ sorted so that $\Delta(\boldsymbol{\tau})_{e_1} \geq \Delta(\boldsymbol{\tau})_{e_2} \geq \ldots \geq \Delta(\boldsymbol{\tau})_{e_{|E|}}$
3:    **for** $i = 0, 1, \ldots, |E|$ **do**
4:       $f_{-i} \leftarrow f(\{e_1, e_2, \ldots, e_{i-1}\})$
5:       $f_{+i} \leftarrow f(\{e_1, e_2, \ldots, e_i\})$
6:       $s_{e_i} \leftarrow f_{+i} - f_{-i}$
7:    **return** $\mathbf{s}$

Figure 2: Inference with Frank-Wolfe, assuming that VAR-INFERENCE guarantees an *upper bound*.

**Projected gradient descent (PGD).** Since it is possible to compute projections onto $B(f)$, and practically so for many submodular functions $f$, we can alternatively use projected gradient or subgradient descent (PGD). Without smoothness, PGD converges at a rate of $O(1/\sqrt{t})$. If the objective is smooth, we can use an accelerated methods like FISTA [25], which has both a much better $O(\sigma/t^2)$ rate and seems to converge faster than many Frank-Wolfe variants in our experiments.

**Smoothness and convergence.** The final question that remains to be answered is under which conditions problem (3) is smooth (the proof can be found in the appendix).

**Theorem 1** *Problem* (3) *is* $k^2\sigma$-*smooth over* $B(f)$ *if the entropy surrogate* $-\overline{H}$ *is* $\frac{1}{\sigma}$-*strongly convex.*

This result follows from the duality between smoothness and strong convexity for convex conjugates, see e.g. [26]. It implies that the convergence rates of the proposed algorithms depend on the strong convexity of the entropy approximation $-\overline{H}$. The benefits of strongly convex entropy approximations are known. For instance, the tree-reweighted entropy approximation is strongly convex with a modulus $\sigma$ depending on the size of the graph; similarly, the SDP relaxation is strongly convex [27]. London et al. [28] provide an even sharper bound for the tree reweighted entropy, and show how one can strong-convexify any weighted entropy by solving a QP over the weights $\boldsymbol{\rho}$.

In practice, because the inner problem is typically solved using an iterative algorithm and because the problem is smooth, we obtain speedups by warm-starting the solver with the solution at the previous iterate. We can moreover easily obtain duality certificates using the results in [24].

**Joint optimization.** When using weighted entropy approximations, it makes sense to optimize over *both* the linearization $\mathbf{g}$ and the weights $\boldsymbol{\rho}$ jointly. Specifically, let $\mathbb{T}$ be some set of weights that yield an entropy approximation $\overline{H}$ that upper bounds $H$. Then, if we expand $\overline{H}$ in problem (3), we obtain

$$\underset{\mathbf{g} \in B(f), \boldsymbol{\rho} \in \mathbb{T}}{\text{minimize}} \sup_{\boldsymbol{\tau} \in \mathbb{L}} \langle \mathtt{stack}(\boldsymbol{\theta}, \mathbf{g}), \boldsymbol{\tau} \rangle + \sum_{i \in V} \rho_i H_i(\boldsymbol{\tau}_i) + \sum_{\{i,j\} \in E} \rho_{i,j} H_{i,j}(\boldsymbol{\tau}_{i,j}).$$

Note that inside the supremum, *both* $\mathbf{g}$ and $\boldsymbol{\rho}$ appear only *linearly*, and there is no summand that has terms from both of them. Thus, the problem is convex in $(\mathbf{g}, \boldsymbol{\rho})$, and we can optimize *jointly* over

both variables. As a final remark, if we already perform inference in a pairwise model and repeatedly tighten the approximation by optimizing over $\boldsymbol{\rho}$ via Frank-Wolfe (as suggested in [19]), then the complexity per iteration remains the same even if we use the higher-order term $f$.

## 4   Submodular lower bounds

While we just derived variational upper bounds, we next develop lower bounds on the partition function. Specifically, analogously to the linearization for the upper bound, if we pick an element $(\mathbf{g}, c)$ of $\mathcal{U}(f)$, the partition function of the resulting pairwise approximation always lower bounds the partition function of (1). Formally,

$$\log \mathcal{Z} \geq \log \sum_{\mathbf{x} \in \{0,1\}^V} \exp \left( - (\mathbf{a}^T \mathbf{x} + \sum_{\{i,j\} \in E} \theta_{ij,x_i x_j} + \sum_{\{i,j\} \in E} g_{i,j} [\![ x_i \neq x_j ]\!] + c) \right) = A(\mathbf{g}) - c.$$

As before, after plugging in a *lower* bound estimate of $A$, we obtain a variational lower bound over the partition function, which takes the form

$$\log \mathcal{Z} \geq \sup_{(\mathbf{g},c) \in \mathcal{U}(f), \boldsymbol{\tau} \in \overline{\mathbb{M}}} -c + \langle \mathtt{stack}(\boldsymbol{\theta}, \mathbf{g}), \boldsymbol{\tau} \rangle + \overline{H}(\boldsymbol{\tau}), \tag{4}$$

for any pair of approximations of $\mathbb{M}$ and $\overline{H}$ that guarantee a lower bound of the pairwise model. We propose to optimize this lower bound in a block-coordinate-wise manner: first with respect to the pseudo-marginals $\boldsymbol{\tau}$ (which amounts to approximate inference in the linearized model), and then with respect to the supergradient $(\mathbf{g}, c) \in \mathcal{U}(f)$. As already noted, this step is in general intractable. However, it is well-known [29] that for any $Y \subseteq E$ we can construct a point (so called *bar supergradient*) in $\mathcal{U}(f)$ as follows. First, define the vectors $\mathbf{a}_{i,j} = f(\mathbf{1}_{\{i,j\}})$ and $\mathbf{b}_{i,j} = f(\mathbf{1}) - f(\mathbf{1} - \mathbf{1}_{\{i,j\}})$. Then, the vector $(\mathbf{g}, c)$ with $\mathbf{g} = \mathbf{b} \odot \mathbf{1}_Y + (\mathbf{1} - \mathbf{1}_Y) \odot \mathbf{a}$ and $c = f(Y) - \mathbf{b}^T \mathbf{1}_Y$ belongs to $\mathcal{U}(f)$, where $\odot$ denotes element-wise multiplication.

**Theorem 2** *Optimizing problem* (4) *for a fixed $\boldsymbol{\tau}$ over all bar supergradients is equal to the following submodular minimization problem* $\min_{Y \subseteq E} f(Y) + \big( \boldsymbol{\Delta}(\boldsymbol{\tau}) \odot (\mathbf{b} - \mathbf{a}) - \mathbf{b} \big)^T \mathbf{1}_Y$.

In contrast to computing the MAP, the above problem has no constraints and can be easily solved using existing algorithms. As the approximation algorithm for the linearized pairwise model, one can always use mean-field [7]. Moreover, if (i) the problem is binary with submodular pairwise potentials $\theta_{i,j}$ and (ii) $f$ is monotone, we can also use belief propagation. This is an implication of the result of Ruozzi [30], who shows that traditional belief-propagation yields a lower bound on the partition function for binary pairwise log-supermodular models. It is easy to see that the above conditions are sufficient for the log-supermodularity of the linearized model, as $\mathbf{g} \geq \mathbf{0}$ when $f$ is monotone (because both $\mathbf{a}$ and $\mathbf{b}$ have non-negative components). Moreover, in this setting both the mean-field and belief propagation objectives (i.e. computing $\boldsymbol{\tau}$) can be cast as an instance of continuous submodular minimization (see e.g. [31]), which means that they can be solved to arbitrary precision in polynomial time. Unfortunately, problem (4) will *not* be jointly submodular, so we still need to use the block-coordinate ascent method we have just outlined.

## 5   Approximate inference via MAP perturbations

For binary models with submodular pairwise potentials and monotone $f$ we can (approximately) solve the MAP problem using the techniques in [1, 4]. Hence, this opens as an alternative approach the perturb-and-MAP method of Papandreou and Yuille [32]. This method relies on a set of tractable first order perturbations: For any $i \in V$ define $\theta_i'(x_i) = \theta_i(x_i) - \eta_{i,x_i}$, where $\boldsymbol{\eta} = (\eta_{i,x_i})_{i \in V, x_i \in \mathcal{X}}$ are a set of independently drawn Gumbel random variables. The optimizer $\mathrm{argmin}_{\mathbf{x}} G_{\boldsymbol{\eta}}(\mathbf{x})$ of the perturbed model energy $G_{\boldsymbol{\eta}}(\mathbf{x}) = \sum_{i \in V} \theta_i'(x_i) + \sum_{\{i,j\} \in E} \theta_{i,j}(x_i, x_j) + f(\mathbf{y}(\mathbf{x}))$ is then a sample from (an approximation to) the true distribution. If this MAP problem can be solved exactly (which is not always the case here), then it is possible to obtain an upper bound on the partition function [33].

## 6   Experiments

**Synthetic experiments.**   Our first set of experiments uses a complete graph on $n$ variables. The unary potentials were sampled as $\theta_i(x_i) \sim \mathrm{Uniform}(-\alpha, \alpha)$. The edges $E$ were randomly split

into five disjoint buckets $E_1, E_2, \ldots, E_5$, and we used $f(\mathbf{y}) = \sum_{j=1}^{5} h_j(\mathbf{y}_{E_j})$, where $\mathbf{y}_{E_i}$ are the coordinates of $\mathbf{y}$ corresponding to that group, and the functions $\{h_j\}$ will be defined below. To perform inference in the linearized pairwise models, we used: `trwbp`, `jtree+` (exact inference, upper bound), `jtree-` (same, lower bound), `sdp` (SDP), `mf` (mean-field), `bp` (belief propagation), `pmap` (perturb-and-MAP with approximate MAP) and `epmap` (perturb-and-MAP with exact MAP). We used `libDAI` [34] and implemented `sdp` using `cvxpy` [35] and `SCS` [36]. As a maxflow solver we used [37]. Errors bars denote three standard errors.

Figure 3 shows the results for $h_i(y_{E_i}) = w_i \sqrt{\sum_{e \in E_i} y_e} / \sqrt{|E_i|}$, with weights $w_i \sim \text{Uniform}(0, \beta)$. In panel (c) we use mixed (attractive and repulsive) pairwise potentials, chosen as $\theta_{i,j}(x_i, x_j) = w_{i,j} [\![ x_i \neq x_j ]\!]$, where $w_{i,j} \sim \text{Uniform}(-\beta, \beta)$. First, the results imply that the methods optimizing the fully convex upper bound yield very good marginal probabilities over a large set of parameter configurations. The estimate of the log-partition function from `trwbp` is also very good, while `sdp` is much worse, which we believe can be attributed to the very loose entropy bound used in the relaxation. The lower bounds (`bp` and `mf`) work well for settings when the pairwise strength $\beta$ is small compared to the unary strength $\alpha$. Otherwise, both the bound and the marginals become worse, while `jtree-` still performs very well. This could be explained by the hardness of the pairwise models obtained after linearizing $f$. Finally, `pmap` (when applicable) seems very promising for small $\beta$.

To better understand the regimes when one should use `trwbp` or `pmap`, we compare their marginal errors in Figure 5. We see that for most parameter configurations, `trwbp` performs better, and significantly so when the edge interactions are strong.

Finally, we evaluate the effects of the approximate MAP solver for `pmap` in Figure 4. To be able to solve the MAP problem exactly (see [4]), we used $h(\mathbf{y}_{E_j}) = \max\{\sum_{e \in E_j} y_e v_e, \sum_{e \in E_j} v_e/2\}$, where $v_e \sim \text{Uniform}(0, \beta)$. As evident from the figure, the gains from the exact solver seem minimal, and it seems that solving the MAP problem approximately does not strongly affect the results.

**An example from computer vision.** To demonstrate the scalability of our method and obtain a better qualitative understanding of the resulting marginals, we ran `trwbp` and `pmap` on a real world image segmentation task. We use the same setting, data and models as [1], as implemented in the `pycoop`[5] package. Because `libDAI` was too slow, we wrote our own TRWBP implementation. Figure 6 shows the results for two specific images (size $305 \times 398$ and $214 \times 320$). The example in the first row is particularly difficult for pairwise models, but the rich higher-order model has no problem capturing the details even in the challenging shaded regions of the image. The second row shows results for two different model parameters. The second model uses a function $f$ that is closer to being linear, while the first one is more curved (see the appendix for details). We observe that `trwbp` requires lower temperature parameters (i.e. relatively larger functions $\theta_i, \theta_{i,j}$ and $f$) than `pmap`, and that the bottleneck of the complete inference procedure is running the `trwbp` updates. In other words, the added complexity from our method is minimal and the runtime is dominated by the message passing updates of TRWBP. Hence, any algorithms that speed up TRWBP (e.g., by parallelization or better message scheduling) will result in a direct improvement on the proposed inference procedure.

## 7 Conclusion

We developed new inference techniques for a new broad family of discrete probabilistic models by exploiting the (indirect) submodularity in the model, and carefully combining it with ideas from classical variational inference in graphical models. The result are inference schemes that optimize rigorous bounds on the partition function. For example, our upper bounds lead to convex variational inference problems. Our experiments indicate the scalability, efficacy and quality of these schemes.

**Acknowledgements.** This research was supported in part by SNSF grant CRSII2_147633, ERC StG 307036, a Microsoft Research Faculty Fellowship, a Google European Doctoral Fellowship, and NSF CAREER 1553284.

## Footnotes

[1]The results presented in this paper can be easily extended to arbitrary binary-valued functions $\mathbf{y}(\mathbf{x})$.

[2]This formulation is slightly nonstandard, but will be very useful for the subsequent discussion in §3.

[3]If we compute the Fenchel dual, we obtain a special case of the problem considered in [22] with the Lovász extension acting as a non-smooth non-local energy function (in the terminology introduced therein).

[4]This is also known as the *diminishing returns property*.

[5] `https://github.com/shelhamer/coop-cut`.

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
