[Supplementary Material]

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

# A Proof of Theorem 1

This result easily follows from well known results in convex analysis, but we give the proof here for completeness. Let $\mathbf{g}$ and $\mathbf{g}'$ be two different linearizations from $B(f)$ with resulting pseudomarginals $\boldsymbol{\tau}$ and $\boldsymbol{\tau}'$ respectively (which are unique, since we assume that the negative entropy is strictly convex). Furthermore, let us denote by $\boldsymbol{\theta}(\mathbf{g})$ (and analogously $\boldsymbol{\theta}(\mathbf{g}')$) the vector of same size as $\boldsymbol{\theta}$ that agrees with $\boldsymbol{\theta}$ on all coordinates except that $\theta(\mathbf{g})_{ij,x_i x_j} = \theta_{ij,x_i x_j} + g_{i,j}$ for $x_i \neq x_j$. Because the negative entropy is assumed to be $\frac{1}{\sigma}$-strongly-convex, we have that

$$\|\boldsymbol{\tau} - \boldsymbol{\tau}'\| \leq \sigma \|\boldsymbol{\theta}(\mathbf{g}) - \boldsymbol{\theta}(\mathbf{g}')\| = \sigma \sqrt{\sum_{\{i,j\} \in E} \sum_{x_i \neq x_j} (g_{i,j} - g'_{i,j})^2} \leq \sigma k \|\mathbf{g} - \mathbf{g}'\|.$$

Then, note that

$$\boldsymbol{\Delta}(\boldsymbol{\tau}) = \underbrace{\begin{pmatrix} \mathbf{1}^T & \mathbf{0}^T & \cdots & \mathbf{0}^T \\ \mathbf{0}^T & \mathbf{1}^T & \cdots & \mathbf{0}^T \\ \vdots & \ddots & \ddots & \vdots \\ \mathbf{0}^T & \mathbf{0}^T & \cdots & \mathbf{1}^T \end{pmatrix}}_{A} \boldsymbol{\tau}_p,$$

where $A$ is a $m \times mk^2$ binary matrix, i.e. each of the blocks $\mathbf{1}^T$ and $\mathbf{0}^T$ are of size $1 \times k^2$, and $\boldsymbol{\tau}_p$ is the vector formed by those coordinates of $\boldsymbol{\tau}$ holding the pairwise marginals. Now, we have that

$$\|\boldsymbol{\Delta}(\boldsymbol{\tau}) - \boldsymbol{\Delta}(\boldsymbol{\tau}')\| = \|A\boldsymbol{\tau}_p - A\boldsymbol{\tau}'_p\| \leq \|A\| \|\boldsymbol{\tau}_p - \boldsymbol{\tau}'_p\| \leq \|A\| \sigma k \|\mathbf{g} - \mathbf{g}'\|.$$

Hence, it remains to compute $\|A\|$. Because $AA^T = k^2 I$, it follows that its top singular value, and thus its norm, is $k$, which completes the proof.

# B Proof of Theorem 2

This result also easily follows by rewriting problem (4) Remember that for a *fixed* $\boldsymbol{\tau}$ we want to maximize

$$-c + \langle \texttt{stack}(\boldsymbol{\theta}, \mathbf{g}), \boldsymbol{\tau} \rangle + \overline{H}(\boldsymbol{\tau}),$$

over $(\mathbf{g}, c) \in \mathcal{U}(f)$. If we expand the stacked part (see eqn. (2)), negate the expression and drop all terms that do not depend on $\mathbf{g}$, we want to *minimize* $c + \boldsymbol{\Delta}(\boldsymbol{\tau})^T \mathbf{g}$. Now, if we replace $\mathbf{g}$ and $c$ by the bar supergradient, we obtain the expression

$$f(Y) - \mathbf{b}^T \mathbf{1}_Y + \boldsymbol{\Delta}(\boldsymbol{\tau})^T (\mathbf{b} \odot \mathbf{1}_Y + (\mathbf{1} - \mathbf{1}_Y) \odot \mathbf{a}).$$

Now the result follows from the following equality

$$\boldsymbol{\Delta}(\boldsymbol{\tau})^T (\mathbf{b} \odot \mathbf{1}_Y + (\mathbf{1} - \mathbf{1}_Y) \odot \mathbf{a}) = \boldsymbol{\Delta}(\boldsymbol{\tau})^T ((\mathbf{b} - \mathbf{a}) \odot \mathbf{1}_Y) + \underbrace{c}_{\text{const. wrt. } Y}$$

$$= \sum_{e \in Y} \Delta(\boldsymbol{\tau})_e (b_e - a_e) + c$$

$$= (\boldsymbol{\Delta}(\boldsymbol{\tau}) \odot (\mathbf{b} - \mathbf{a}))^T \mathbf{1}_Y + c.$$

# C Experiments

## C.1 Synthetic experiments

Below we provide the configuration file that was used for the experiments. The parameters inside `params_inf` were what was used to initialize the `libDAI` inference methods. The parameters inside `params_coop` specify the algorithm to be used for optimization over $B(f)$. We are using FISTA with a fixed step size of `eta` and convergence tolerance of `eps`.

```
{
    "trwbp": {                                          "bp": {
        "class_name": "DAIInference",                       "class_name": "DAIInference",
        "params_inf": {                                     "params_inf": {
            "name": "TRWBP",                                    "name": "BP",
            "verbose": 0,                                       "verbose": 0,
            "tol": 0.001,                                       "tol": 0.001,
            "logdomain": 1,                                     "logdomain": 1,
            "updates": "SEQFIX",                                "updates": "SEQFIX",
            "inference": "SUMPROD",                             "inference": "SUMPROD"
            "nrtrees": 1000                                 },
        },                                                  "params_lower": {
        "params_coop": {                                    }
            "eps": 0.001,                               },
            "max_iters": 100,                           "mf": {
            "method": "fista",                              "class_name": "DAIInference",
            "variant": "fixed",                             "params_inf": {
            "eta": 1                                            "name": "MF",
        }                                                       "verbose": 0,
    },                                                          "tol": 0.001,
    "jtree+": {                                                 "logdomain": 1,
        "class_name": "DAIInference",                           "updates": "NAIVE",
        "params_inf": {                                         "inference": "SUMPROD",
            "name": "JTREE",                                    "maxiter": 10000
            "updates": "HUGIN"                              },
        },                                                  "params_lower": {
        "params_coop": {                                    }
            "eps": 0.001,                               },
            "max_iters": 100,                           "sdp": {
            "method": "fista",                              "class_name": "SDPInference",
            "variant": "fixed",                             "params_inf": {
            "eta": 1                                            "solver": "SCS",
        }                                                       "warm_start": true
    },                                                      },
    "jtree-": {                                             "params_coop": {
        "class_name": "DAIInference",                           "eps": 0.001,
        "params_inf": {                                         "max_iters": 100,
            "name": "JTREE",                                    "method": "fista",
            "updates": "HUGIN"                                  "variant": "fixed",
        },                                                      "eta": 1
        "params_lower": {                                   }
        }                                               }
    },                                              }
```

## C.2 Image segmentation

The model of Jegelka and Bilmes [1] has two parameters: $\lambda$, which controls the strength of the unaries, and $\theta$, which controls the curvature of the function $f$. In addition to these parameters, we have another parameter $t$ that controls the inverse temperature of the model, i.e. the complete energy in (1) is multiplied by $t$. In the table below we provide all parameters for Figure 6.

| Panel | $\theta$ | $\lambda$ | $t$ |
|-------|-------|---|-----|
| (b) | 1 | 1 | 0.1 |
| (c) | 1 | 1 | 0.1 |
| (d) | 0.1 | 1 | 0.1 |
| (e) | 0.001 | 1 | 0.1 |
| (g) | 0.01 | 1 | 1 |
| (h) | 0.01 | 1 | 1 |
| (i) | 0.1 | 1 | 0.1 |
| (j) | 0.1 | 1 | 0.1 |