[Reviews · NeurIPS 2016]

Reviewer 1

Summary

This is an interesting paper on providing upper and lower bounds on Markov random field models that include an additional, non-factored but submodular energy term. It uses two existing tricks: (1) upper and lower linear bounds on the submodular function f(.), which convert f(.) to additional low-order factor terms, and (2) standard variational upper and lower bounds on the resulting MRF. In combination, these give upper and lower bounds on the original MRF's partition function.

Qualitative Assessment

This is an interesting paper on providing upper and lower bounds on Markov random field models that include an additional, non-factored but submodular energy term. It uses two existing tricks: (1) upper and lower linear bounds on the submodular function f(.), which convert f(.) to additional low-order factor terms, and (2) standard variational upper and lower bounds on the resulting MRF. In combination, these give upper and lower bounds on the original MRF's partition function. It's not entirely clear to me how useful these resulting bounds are; in the experiments, the authors mainly report the pseudomarginals of the upper bound, rather than actually doing anything with the partition function. I would expect the upper bounds to be mostly helpful in parameter learning problems. It would be nice if the authors could comment on this. Connections to existing work are very good and clearly explained. On a technical point, I would suggest replacing the marginal polytope form of TRW in (3) with a decomposition form (the dual of the form in (3)), for example a combination of spanning trees (Jancsary & Matz, AISTATS 2011), or dual decomposition (Ping et al., NIPS 2015). This would have the advantage of converting the saddle point, double-loop optimization into a single joint minimization, simplify the weight updates, and likely speed the whole thing up since your computation is dominated by running TRWBP in the inner loop.

Confidence in this Review

2-Confident (read it all; understood it all reasonably well)


Reviewer 2

Summary

Authors adopt the “cooperative graph cut” approach of Jegelka and Bilmes and present an approach for approximate probabilistic inference, which is more involved than with basic pairwise graphical models due to the nonlocal “cooperative term” f in (1). The basic idea is to linearize this problematic term and to adapt it through the outer optimization task of (3), based on outer convex relaxations for lower bounding the log-partition function and on non-convex inner relaxations (mean-field) for the upper bound. Since probabilistic inference (as opposed to MAP) entails the corresponding surrogate of the entropy, the dependency of (3) on g is smooth (i.e. C^{1,L}) and hence there is no need to resort to inefficient subgradient-based schemes.

Qualitative Assessment

The novelty of the paper concerns the knowledgeable assembly of established components. The resulting overall approach should find many applications. The title sounds too general to me in view of related work, see e.g. http://dx.doi.org/10.1007/s10107-016-1038-y and related prior work. More appropriate would be something like: Efficient probabilistic inference with cooperative graph cuts.

 The presentation lacks precision here and there, to some extent to lack of space, of course. I pick out three points and ask authors to clarify and to improve the presentation: (i) while the interplay between convexity and smoothness through duality is clear, a precise reference to a quantitative statement in the literature would be appropriate, in connection with theorem 1. (ii) Strong convexity of a functions means that subtraction of the Euclidean squared norm (multiplied with some fixed constant) does preserve convexity. Since this is not the case e.h. for the basic entropy function \sum_i x_i log(x_i), it is not immediate and obvious that global strong convexity holds for -\bar{H} (theorem 1). The claimed applicability of FISTA requires a global Lipschitz constant too. Please specify this point in view of the bounded feasible sets. (iii) If the quantitative assertion of Thm. 1 holds, then the Lipschitz constant depends on the number of labels $k$. In realistic application, this number can be much larger than in your toy experiments and hence should slow down optimization. Please comment.

Confidence in this Review

2-Confident (read it all; understood it all reasonably well)


Reviewer 3

Summary

In this paper, the authors introduce a new class of probabilistic models (cooperative graphical model) and derive the corresponding variational inference schemes. The cooperative model is defined by adding an extra submodular function to energy of pairwise graphical models. This terms allows for higher-order interactions between the variables, compared to low-treewidth or pairwise models. The authors take advantage of the submodular structure and design efficient optimization algorithms to obtain both lower and upper variational bounds for the partition function. Numerical experiments show the quality of the bounds and the advantage of this cooperative graphical model on a small scale segmentation task.

Qualitative Assessment

pros 1. Cooperative graphical models are formulated clearly and concisely. 2. This paper demonstrates the process of combining traditional variational inference methods with the two polytopes associated with the submodular function f to compute convex upper bounds (base polytope) and submodular lower bounds (upper submodular polyhedron) 3. Details are well illustrated when solving intermediate optimization problems. 4. Experimental results are convincing and well analyzed to show the effectiveness and efficiency of the various optimization approaches. cons: 1. The novelty with respect to [11] is a bit limited. It's true that this family of models generalizes the ones in [11], but the variational derivation using the base polytope B(f) is quite similar to [11]. 2. The core idea of cooperative graphical models is the extra submodular term. The main message of the paper is saying that this extra term will still allow for varaitional inference algorithms (exact inference is intractable, because it generalizes Ising models). The significance of this work is largely determined by the extra modeling power obtained by introducing the submodular term in the potential function. More experiments showing the advantage of this extra term would be desirable. 3. The paper develops tractable optimization algorithms for the new cooperative framework. The extra submodular term gives better experimental performance (as in Figure 6.), however, the optimization is more expensive. It would be good to report runtimes to better compare the traditional pairwise model and the cooperative model.

Confidence in this Review

2-Confident (read it all; understood it all reasonably well)


Reviewer 4

Summary

The authors develop methods for approximate marginal inference and partition function estimation in an interesting and useful class of 'cooperative graphical models' where an extra submodular cost function on edges significantly extends pairwise MRFs in ways that can be helpful e.g. for image segmentation. Earlier work considered MAP inference for a subclass of these models but here marginal inference is tackled by cleverly combining polyhedral results with earlier work on variational inference.

Qualitative Assessment

The paper is interesting, clear and well written with good motivation, background and examples. Novel upper bounds (convex) and lower bounds are derived for log Z, then a good range of existing inference methods are used appropriately to estimate these bounds, including Frank-Wolfe, PGD, TRW, BP, mean field and perturb-and-MAP methods. These are evaluated empirically and initial conclusions and guidance are drawn. For the empirical results on small models, how was exact inference performed - I presume brute force? The larger example from computer vision is helpful and demonstrates the promise of this approach. Lines 216-223: Nice idea to use Bethe here 224: nice again but don't we again need both conditions from just above, (i) and (ii) for this? Minor points: In Abstract, perhaps clarify: efficient inference techniques -> methods for approximate inference (clarifying approximate and also perhaps efficient should not strictly be used since eg BP does not have guaranteed runtime) Footnote 1: could this be elaborated in the Appendix Line 114: for clarity, perhaps something like "with an eye toward the bounds of Sec 3, we write a model using g instead of f, as..." Equation after line 29: typo mu-> tau 146: an entropy -> a concave entropy 236: solved exactly -> solved exactly or upper bounded

Confidence in this Review

2-Confident (read it all; understood it all reasonably well)


Reviewer 5

Summary

This paper provides variational lower/upper bound of partition of a special class of graphical models called `cooperative graphical model’. Cooperative graphical model is a probability distribution which is similar with pairwise graphical models but its energy function contains some extra submodular function of disagreement variable. The main technique for formulating variational lower/upper bound of cooperative graphical model is finding pairwise graphical model and its associate polytope which upper/lower bound the original cooperative graphical model. The author also provides algorithms for suggested bounds and experimental results for image segmentation.

Qualitative Assessment

I think the proposed graphical model is interesting and novel. The authors provide a lower bound and a fully convex upper bound for partition function. In particular, variational upper/lower bound are formulated by using pairwise graphical models, e.g. mean field approximation and SDP, and therefore, one can apply existing variational methods for pairwise graphical models for cooperative graphical models. The author also provides algorithms, Frank-Wolfe, PGD and BP, for finding upper/lower bound. The experimental results shows that cooperative graphical models and its variational methods performs better than pairwise graphical models for image segmentation. Variational bounds provided in the paper are formulated by bounding the cooperative graphical model by pairwise models and use the existing variational bounds for pairwise models to bound cooperative graphical model. However, as the author applies two bounds consecutively, the suggested bound might be loose. Lower bound is not tractable. Since the optimization over a polytope U(f) is not tractable, the variational lower bound is also intractable. The author suggests an algorithm based on the block coordinate descent which finds the lower bound for the suggested variational lower bound, however, the algorithm only explore the finite points of U(f) and it may not provide a tight lower bound. The technique of constructing B(f) and U(f) seems generic. Can this technique applied to a wider class of f or other probability distribution? Furthermore, the optimization objectives (3), (4) for the variational upper/lower bound does not seems to be dependent with the submodularity of f (while the algorithms do). Typos - In page 2, line 59~60, for all x,y -> for all y,y^\prime - In page 5, procedure Linear-Oracle, `)’ is missing for procedure 4,5 - In page 6, line 211, U(F) -> U(f) - In page 5, line 179, smoth -> smooth

Confidence in this Review

2-Confident (read it all; understood it all reasonably well)